# High-Frequency Magnetoimpedance (MI) and Stress-MI in Amorphous Microwires with Different Anisotropies

**DOI:** 10.3390/nano11051208

**Published:** 2021-05-02

**Authors:** Junaid Alam, Makhsudsho Nematov, Nikolay Yudanov, Svetlana Podgornaya, Larissa Panina

**Affiliations:** 1Institute of Novel Materials and Nanotechnology, National University of Science and Technology, MISiS, 119991 Moscow, Russia; engr.sajalam@gmail.com (J.A.); kolyan2606@mail.ru (N.Y.); podgsv@mail.ru (S.P.); 2Institute of Physics, Mathematics & IT, Immanuel Kant Baltic Federal University, 236041 Kaliningrad, Russia; nmg2409@gmail.com; 3Faculty of Energy, Tajik Technical University Named after ac. M.S. Osimi, Dushanbe 734042, Tajikistan

**Keywords:** amorphous microwires, high-frequency magnetoimpedance, SOLT calibration, magnetic anisotropy

## Abstract

Magnetoimpedance (MI) in Co-based microwires with an amorphous and partially crystalline state was investigated at elevated frequencies (up to several GHz), with particular attention paid to the influence of tensile stress on the MI behavior, which is called stress-MI. Two mechanisms of MI sensitivity related to the DC magnetization re-orientation and AC permeability dispersion were discussed. Remarkable sensitivity of impedance changes with respect to applied tensile stress at GHz frequencies was obtained in partially crystalline wires subjected to current annealing. Increasing the annealing current enhanced the axial easy anisotropy of a magnetoelastic origin, which made it possible to increase the frequency of large stress-MI: for 90mA-annealed wire, the impedance at 2 GHz increased by about 300% when a stress of 450 MPa was applied. Potential applications included sensing elements in stretchable substrates for flexible electronics, wireless sensors, and tunable smart materials. For reliable microwave measurements, an improved SOLT (short-open-load-thru) calibration technique was developed that required specially designed strip cells as wire holders. The method made it possible to precisely measure the impedance characteristics of individual wires, which can be further employed to characterize the microwave scattering at wire inclusions used as composites fillers.

## 1. Introduction

Soft ferromagnetic microwires have attracted growing attention in research since they exhibit a number of physical effects important for applications (see, for example, recent reviews [1,2,3]). Controlling their microstructure, chemical composition, and phase composition, cross-sectional size allows for the development of novel concepts for their applications in sensory devices and functional materials. Here, we investigated high frequency magnetoimpedance (MI) in Co-based microwires with amorphous and partially crystalline states, paying particular attention to the influence of a tensile stress on the MI behavior at elevated frequencies. Potential applications include sensing elements in stretchable substrates for flexible electronics, wireless sensors, and tunable smart materials [4,5,6].

The MI effect, which is referred to as a large and sensitive change in the complex-valued impedance of a ferromagnetic conductor in the presence of external magnetic field and other stimuli such as a mechanical stress and temperature, led to the development of various sensors driven by alternative current (AC) or pulse circuits operating at megahertz frequencies [7,8,9,10,11]. A high sensitivity of MI with respect to the external factors is caused by the directional change in the quasi-static (DC) magnetization. Therefore, it was expected that MI remains sensitive for higher frequencies in the GHz range. At high frequencies, when the skin effect is strong, the impedance dependence on the magnetic parameters has the form [12]:(1)$$Z=Z_{c}\left(\sqrt{\mu_{ef}}\cos^{2}\theta +sin^{2}\theta \right)$$

Here, $Z_{c}$ is the impedance of a non-ferromagnetic conductor, $\mu_{ef}$ is the AC permeability parameter, and θ is the angle between the static magnetization and high-frequency current. Equation (1) shows that as long as the angle θ varies in response to external factors and the value of $\mu_{ef}$ differs from unity, the high frequency impedance demonstrates $\cos^{2}\theta$-dependence. The permeability also depends on θ and other DC parameters, but this dependence is typically weak since high values of a DC magnetic field and/or anisotropy are required to satisfy the condition of the ferromagnetic resonance at these frequencies [13]. A typical magnetic field behavior of impedance at GHz frequencies in Co-rich amorphous microwires with a well-formed circumferential easy anisotropy is given in Figure 1. The inset shows the DC magnetization loop, and thus it is clearly seen that the field interval where the magnetization rotates towards the axis corresponded to a sharp increase in impedance. When the static magnetization was along the wire (θ=0) and did not change by the field, only the permeability parameter $\mu_{ef}\left(H\right)$ determined the field behavior of the impedance: it increased with much lower slope, and an inflexion point was seen in the plot of Z(H). In this field range, the sensitivity of the impedance change was low, of about a few percent per Oe. Similar high-frequency impedance behaviors in these types of wires were demonstrated in a number of publications [14,15,16].

To enhance the MI sensitivity, researchers have proposed various annealing regimes in order to establish a required transverse anisotropy of small amplitude and small easy-axes deviations [17,18,19,20,21]. In the case of stress-MI, the induced circumferential anisotropy in combination with positive magnetostriction formed by current annealing leads to a large change in impedance in response to tensile stress without use of a bias magnetic field [21,22]. This effect is of high practical importance; however, the impedance vs. stress sensitivity can quickly decrease with increasing frequency. Sensitive change in the DC magnetization requires a small anisotropy; then, the ferromagnetic frequency is low, and the frequency range of hundred MHz corresponds to the tail of the ferromagnetic resonance. As follows from Equation (1), if the values of $\mu_{ef}$ are close to unity, the DC magnetic configuration does not affect the impedance.

An interesting situation can occur in wires with a non-uniform-induced anisotropy or with a relatively high axial anisotropy. Thus, large variations of impedance under the effect of mechanical stress can be expected if it causes the anisotropy regions redistribution. In the case of high axial anisotropy of magnetoelastic origin, the ferromagnetic resonance frequency is in the GHz range, with the external stress changes this frequency causing large changes in stress-MI. Here, we investigated the effect of tensile stress on high-frequency impedance in wires with different anisotropy and magnetostriction modified by current annealing. It was found that modifications in the axial anisotropy caused by applied stress led to a large impedance change at high frequencies due to shift in the resonance frequency.

To optimize the impedance behavior, one must measure the impedance characteristics of individual wires that may present considerable problems at higher frequencies in the GHz range. This is related to uncertainties occurring due to calibration of measuring devices (network analyzer) with customized sample arrangements. In this case, customized calibration methods should be used. Coaxial and microstrip transmission lines are widely used as sample holders [23,24]; however, in the case of the impedance measurements when the sample is subjected to mechanical stress, coaxial lines are not convenient. Here, we discuss the calibration procedures for microstrip lines. In the microwave de-embedding methods, the electrical reference planes are shifted to the sample location. This requires the use of dummy sample holders with specific terminations, for example, three reflection standards (open, short, and match load) and two ports connected together (thru). This calibration procedure is known by the name of short-open-load-thru (SOLT) or thru-open-short-match (TOSM) [25]. Accurate calibration standards can be fabricated on the customized printed circuit board (PCB), and then the complex impedance *Z* is calculated from *S*_21_ parameter using a simple formula without solving any equations.

The accuracy of the customized calibration methods critically depends on the tolerance of calibration standards fabrication (for example, the lumped open, short, etc., terminations) and the problem becomes more essential with increasing frequency. Relating to this, an alternative of VNA calibration technique for MI measurements at a high frequency was recently introduced in [26] using zero-field de-embedding method. In this method, the MI measurements obtained at zero magnetic field were used as a reference signal, and then the same process was repeated with applied magnetic field so that the data could be subtracted by the values of the reference signal. However, the major disadvantage of this method is the absence of zero-field behavior. In this work, the SOLT calibration standard was adjusted by using specially designed strip cells as a PCB SOLT standard for VNA calibration and measuring the high frequency MI and stress-MI, which could be directly used for the development of various high-performance micro-sensor devices.

## 2. Materials and Methods

### 2.1. Wire Samples

Glass-coated amorphous microwires of two Co-based compositions Co_66.6_Fe_4.28_B_11.51_Si_14.48_Ni_1.44_Mo_1.69_ and Co_71_Fe_5_B_11_Si_10_Cr_3_ produced by modified Taylor–Ulitovskiy method [27,28] were used in this work. The wires of the first composition had a total diameter D=25.8 µm and a metal core diameter d=14.2 µm (referred to as sample No1). The wire samples of the second composition had different geometry: D=30 µm, d=25 µm and D=43 µm, d=35.2 µm (referred to as samples No2 and No3, respectively).

The wires with smaller diameter were fully amorphous, whereas the wires with larger diameter (sample No 3) were partially crystallized, as was concluded from DSC (differential scanning calorimetry) analysis in previous works [29,30]. Co-based wires typically have a negative magnetostriction and a circular type of the easy anisotropy since the internal stress is predominantly axial. Samples No 1 with a larger ratio of Co/Fe~15 content had the magnetostriction constant of about $-1.2\cdot 10^{-7}$ [30,31] and a circumferential easy anisotropy. Sample No2 had a smaller negative magnetostriction (<$10^{-7}$ [21,32]) but an axial easy anisotropy. Sample No3 had a large and positive magnetostriction constant of about $10^{-6}$ due to partial crystallization [32] and an axial easy anisotropy. The crystallization and Curie temperatures of the studied samples determined from the DSC curves using standard software application were about 790 and 720–730 K, respectively.

The wire samples were annealed by DC current in air atmosphere with the aim to investigate the effect of anisotropy and magnetostriction change on the MI and stress-MI behavior. The sample length for annealing was 15 cm, the DC current intensity was chosen between 25–90 mA, and the flowing current time was 30 min. All current treatments were performed in the same ambient conditions. The current magnitude was chosen to realize a heating effect in the range of temperatures 450–750 K. For moderate annealing currents, the annealing temperature was measured using the temperature dependence of resistivity in a bridge circuit. For higher annealing currents (>60 mA), the annealing temperature was estimated from modelling on the basis of the energy balance. The details are given in [21].

### 2.2. DC Magnetic Measurements

The magnetic hysteresis loops were determined by an inductive method utilizing two differential detection coils with 3 mm in the inner diameter and 5 cm in length. The magnetizing field in the direction of the wire length (axial direction) with a frequency of 500 Hz and amplitude of up to 1000 A/m was sufficient for re-magnetizing the soft magnetic wires. To increase the accuracy of the magnetization loop determination, we used digital integration of the induced voltage.

### 2.3. Impedance Measurements

To obtain reliable measurement results at high frequencies by SOLT calibration, we used specially designed strip cells, shown in Figure 2, as calibration standards. The calibration process is the same as for standard SOLT/TOSM [25,33], but instead of coaxial standards, we used customized strip cells (PCB board). The microwire was installed in the measuring cell of PCB. This calibration technique allows one to move the reference planes towards the ends of the wire sample and only measure its S-parameters.

For the impedance measurements, the standards were mounted on PCB as demonstrated in Figure 3. The PCB calibration and measuring cells utilized FR4 material (glass-reinforced epoxy laminate) with a relative permittivity of 4.35 for frequencies higher than 500 MHz. It was assumed that $S_{11,\;open}=1,\;S_{11,\;short}=-1$, $S_{11,\;load}=0$ for any frequencies. For THRU, a direct connection of the signal stripes was used, then $S_{21,\;thru}=S_{12,\;thru}=1,\;S_{11,\;thru}=0,\;S_{22,\;thru}=0$.

In the microwave region, the length of electromagnetic waves along the sample may become of the order of its length, and thus the sample impedance would no longer be regarded as lumped, and hence the deduced impedance value will contain a significant error. The correction can be made considering that the sample placed between the stripes above the dielectric with the continuous ground on its opposite side represents a planar waveguide, as shown in Figure 3 [16]. The difference from the usual strip line construction is only due to the geometry of a conductor: a wire instead of a strip.

Therefore, this wire waveguide can be characterized by standard parameters: losses and delay time. The losses must be attributed to the intrinsic physical properties of the sample. On the contrary, the delay time ∆*t* is caused mostly by the length of the sample. However, it introduces significant phase distortions at higher frequencies:(2)$$S_{21}\left(\omega \right)={\tilde{S}}_{21}\exp\left(j\gamma \left(\omega \right)\right)\exp\left(-j\omega ∆t\right)$$
where ${\tilde{S}}_{21}$ is the wire lamped $S_{21}$ parameter, ω=2πf is the angular frequency, j is the imaginary unit, γ(ω) is a phase function contributing to internal impedance properties, and exp(−iω∆t) is responsible for the phase incursion due to the time delay ∆t. Thus, after measuring ∆t along the sample, its intrinsic ${\tilde{S}}_{21}$ parameter can be recovered from the measured one by multiplying by exp(jω∆t).

This parameter is already lumped from the point of view of the propagating wave, although it depends on the length of the sample. Using Equation (2), we can calculate the value of impedance, free of the waveguide properties, as
(3)$$Z\left(\omega ,H\right)=\frac{100\;\cdot \;(1-{\tilde{S}}_{21}\left(\omega ,H\right))}{{\tilde{S}}_{21}\left(\omega ,H\right)}$$

Figure 4 shows an example of the impedance dispersion of an amorphous microwire, measured with the modified SOLT calibration (user’s ideal calibration kit created on VNA).

With increasing the frequency, the real part of the impedance starts to decrease, becoming negative, while the imaginary part grows up, becoming positive. This non-physical behavior is caused by the time delay. For removing this effect, we applied an additional adjustment described above. The impedance spectra after phase compensation are shown in Figure 5. The resonance seen at a frequency of about 5 GHz is a parasitic electric resonance related to the OPEN terminations on the PCB calibration cell. It cannot be removed, and a sort of smooth interpolation is needed in this frequency range.

The parasitic resonances observed between 5 and 6 GHz were introduced during the SOLT calibration and were caused by the PCB cell and connection lines. These resonances are well localized and can be easily replaced with smooth interpolants.

The Hewlett-Packard 8753E Vector Network Analyzer was used to measure the wire impedance in the frequency range between 0.1 and 2.5 GHz at room temperature. The wire length for impedance measurements was 11 mm. The impedance spectra were deduced from *S*_21_ parameter (two-port measuring scheme) using Equation (3). The sample was placed inside a Helmholtz coil that produced a slowly varying magnetic field up to 50 Oe. In order to introduce a tensile stress during *S*_21_ measurement, we hanged the load with a thread in the middle part of the wire. The load weight was applied to the whole wire, and thus the stress in the metallic core should be evaluated considering different Young’s moduli of the metal and glass [34].

## 3. Results and Discussions

### 3.1. DC Properties

We investigated the wires of two Co-based compositions with very similar relative content of Co and Fe: 15.6 and 14.2, respectively. It is known that if this ratio is larger than 14, the magnetostriction is negative [31]. Therefore, it could be expected that all the samples have a negative magnetostriction, a circumferential easy anisotropy, and inclined hysteresis loops. However, only sample No1 showed an expected hysteresis loop in as-cast state as shown in Figure 6.

Samples No2 and No3 had rectangular hysteresis loops, which is associated with an axial easy anisotropy. In the case of sample No2, the magnetostriction constant was negative but very small, in the order of −10^−7^ [21]. The axial anisotropy could have been due to the existence of residual Co crystals with large magnetic anisotropy. In the case of sample No3, its magnetostriction was relatively large and positive (approximately ≈10^−6^ [32]) due to partial crystallization, and thus the loop had a rectangular shape. The loop appeared to be asymmetrical, but this was related to digital integration of a sharp voltage pulse induced during re-magnetization. In all the cases, we used current annealing to modify the anisotropy and magnetostriction. It is known that current annealing of amorphous wires when heating below the Curie temperature induces a circumferential easy anisotropy [35]. This was observed in wires with small magnetostriction, but in the case of high and positive magnetostriction, the induced circumferential anisotropy may be insufficient to overcome the magnetoelastic anisotropy and the easy axis of anisotropy would remain axial. In all the cases, higher annealing currents producing the heating above the Curie temperature did not form an induced circumferential anisotropy, but heating increased the magnetostriction constant. This strengthened the axial easy anisotropy. The results of the change in the hysteresis loops due to annealing with different current *I*_an_ are shown in Figure 7.

### 3.2. Impedance vs. Field at High Frequencies

With this choice of wire samples, we had different combinations of easy anisotropy and magnetostriction. First, we examined the impedance frequency behavior. For as-cast sample No1, the frequency plots are shown in Figure 1 and in Figure 8a for higher frequencies. In this case, large change in impedance due to magnetization re-orientation was observed for frequencies higher than 2 GHz, and the corresponding sensitivities were 42%/Oe at 1.5 GHz and 26.6%/Oe at 2.1 GHz.

In this case, current annealing led to a combination of induced circumferential anisotropy and positive magnetostriction, which formed a helical type of easy anisotropy. The change in magnetostriction after annealing was due to structural relaxation and modification in atomic co-ordination [21,36]. The variation of impedance at low fields due to magnetization re-orientation was smaller and quickly decreased with increasing frequency, as seen in Figure 8b,c.

For a wire with an axial anisotropy (Figure 9, sample No2) the impedance vs. field dependence was only due to the permeability dispersion (see Equation (1)), and there was no sensitive impedance change at low frequencies where the MI plots had a single peak at zero field. With increasing frequency, two symmetrical peaks appeared and the field at maximum increased with increasing a frequency due to increase in the frequency of the ferromagnetic resonance. Nevertheless, at a frequency of about 1 GHz, the relative change in impedance ($∆Z/Z_{0}=\left|\left(Z\left(H\right)-Z_{0}\right)/Z_{0}\right|$) $Z_{0}=Z\left(H=0\right)$ was about 10%/Oe. It was also noticed that in the case of an axial anisotropy, the impedance field sensitivity was higher for higher frequencies and remained at a level of 5–7%/Oe up to 1.5 GHz whilst the maximum sensitivity at 100 MHz was only 4.5%/Oe. Similar impedance characteristics were observed for sample No3, which also had an axial easy anisotropy. This behavior suggests that in the case of an axial anisotropy, the change in the anisotropy value may strongly influence the impedance at elevated frequencies.

After optimal current annealing, sample No2 had an induced circumferential anisotropy and positive magnetostriction. The impedance spectra are shown in Figure 10. The sensitive low-field region appeared and the maximal sensitivity at 1 GHz was about 50%/Oe and about 15%/Oe for a frequency of 1.5 GHz. These values were smaller than for sample No1 in as-prepared state since the induced anisotropy was of a helical type and its value was slightly lower. However, the combination of circumferential (or helical anisotropy) and positive magnetostriction was of interest to realize large impedance change at zero field [20,21], as is discussed in the next section.

### 3.3. Effect of Tensile Stress on Impedance

In the case of a sample with a circumferential anisotropy and negative magnetostriction, the application of a tensile stress resulted in increase in the circumferential anisotropy, and thus the main effect widened the impedance plots, as shown in Figure 11.

In this case, large impedance changes in wires subjected to tensile stress requires the application of a DC bias field.

For amorphous wires with the combination of axial anisotropy and negative magnetostriction (sample No2, as-prepared) and circumferential anisotropy and positive magnetostriction (sample No2, current annealed), the tensile stress affected the impedance even at zero magnetic field due to a change in the magnetization direction by stress, as shown in Figure 12.

The strongest change in impedance caused by the application of tensile stress was observed for current annealed wire, which had a combination of induced circular anisotropy and positive magnetostriction. A large stress-MI effect preserved up to 1 GHz, as shown in Figure 13. For a range of 300 MPa, the impedance increased almost four times at a frequency of 0.5 GHz, and it still increased by 60% at a frequency of 1.5 GHz.

In the case of an axial anisotropy and positive magnetostriction, the effect of tensile stress on MI was related to the anisotropy change and its influence on the permeability frequency dispersion. Unexpectedly, this could be even stronger than the effect of magnetization re-orientation. At low frequencies, the impedance values decreased with increasing an axial anisotropy (that is, with increasing a tensile stress) since it was mainly determined by the behavior of the initial rotational permeability, which was inversely proportional to the anisotropy field. With increasing frequency, higher anisotropy better suited the conditions for ferromagnetic resonance, and the permeability increased with increasing anisotropy. In the intermediate region, there was a non-monotonic dependence of impedance on anisotropy (and stress). The evolution of MI behavior under stress for sample No3 with an axial anisotropy is shown in Figure 14.

Axial magnetoelastic anisotropy for sample No3 was increased by increasing the annealing current, as follows from the impedance behavior at low frequencies (<100 MHz): the impedance value at zero field decreased with increasing $I_{an}$. This could have been due to increase in the magnetostriction constant by annealing. For $I_{an}=90\;\mathrm{mA}$, a monotonic increase in the impedance (at zero field) was observed up to frequencies of few GHz when a tensile stress was applied, as shown in Figure 15.

Figure 16 compares the zero-field impedance vs. stress for sample No3 after different annealing conditions.

It is seen that with increasing the annealing current, the impedance stress sensitivity dropped at lower frequencies, but it increased at higher frequencies. Thus, for a wire annealed with a current of 90 mA, the impedance increased by about 300% when a stress of 450 MPa was applied.

## 4. Conclusions

In this work, the MI effect at elevated frequencies was compared for Co-based microwires with different anisotropy and magnetostriction. To realize various combinations of the easy anisotropy and magnetostriction, we successfully used the current annealing treatment of amorphous and partially crystalline wires. Two mechanisms of large MI sensitivity at GHz frequencies were demonstrated: (i) due to DC magnetization re-orientation and (ii) due to anisotropy-dependent permeability dispersion. The formation of a large magnetoelastic anisotropy with an axial easy axis in current annealed Co-based microwires with partial crystallization resulted in large and sensitive stress-MI at GHz frequencies. The latter has a potential for developing stress-sensitive elements, especially for wireless operation at microwave frequencies.

The modified method of SOLT calibration with specially designed strip cells on PCB board could allow for the customized and precise impedance measurements at GHz frequency range, which is quite useful for designing various remote sensors and sensing materials operating at microwave band.

## Figures and Tables

**Figure 1 nanomaterials-11-01208-f001:**
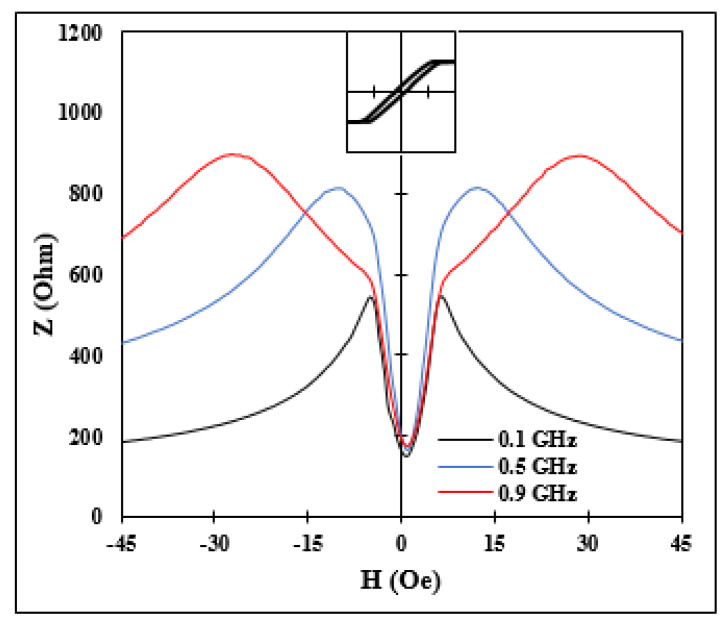
Typical MI in wires with circumferential anisotropy. For this measurement, we used amorphous Co_66.6_Fe_4.28_B_11.51_Si_14.48_Ni_1.44_Mo_1.69_ microwires. Inset shows the DC loop along the wire.

**Figure 2 nanomaterials-11-01208-f002:**
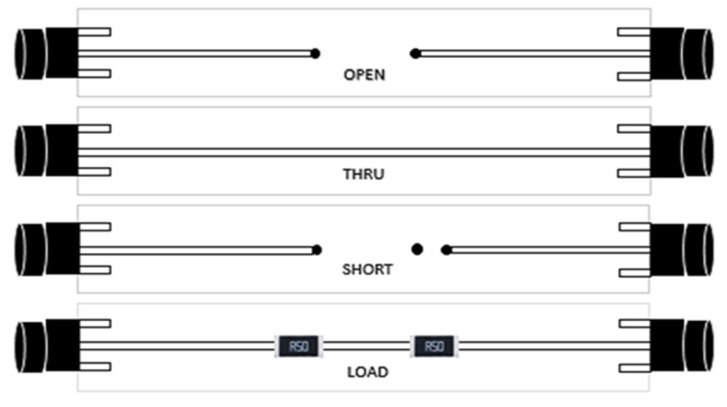
Schematic of specially designed PCB calibration cells with the SOLT standards. THRU standard utilizes direct connection (“flush THRU”) with $S_{21}=S_{12}=1,\;S_{11}=0,\;S_{22}=0$; SHORT is realized via connecting the signal strip to the ground; LOAD utilizes two 50 Ω RF Vishay resistors connected to the ground in parallel.

**Figure 3 nanomaterials-11-01208-f003:**
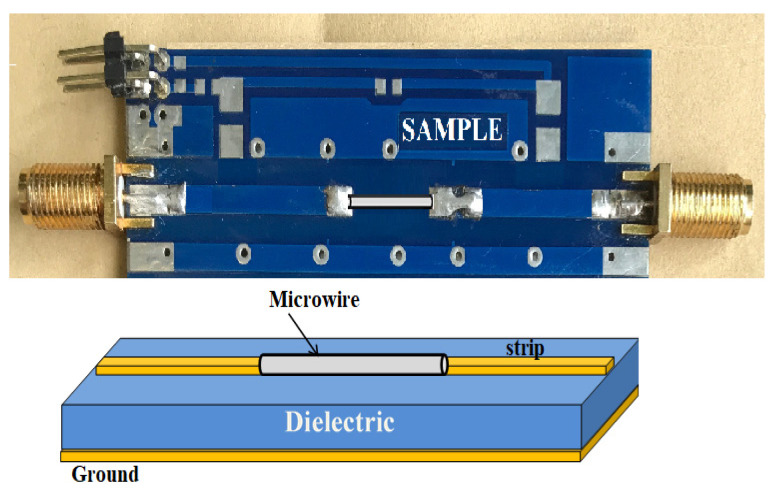
Wire sample as a planar waveguide resulting in a phase incursion: upper figure is a sample holder photo; bottom figure represents a schematic illustration.

**Figure 4 nanomaterials-11-01208-f004:**
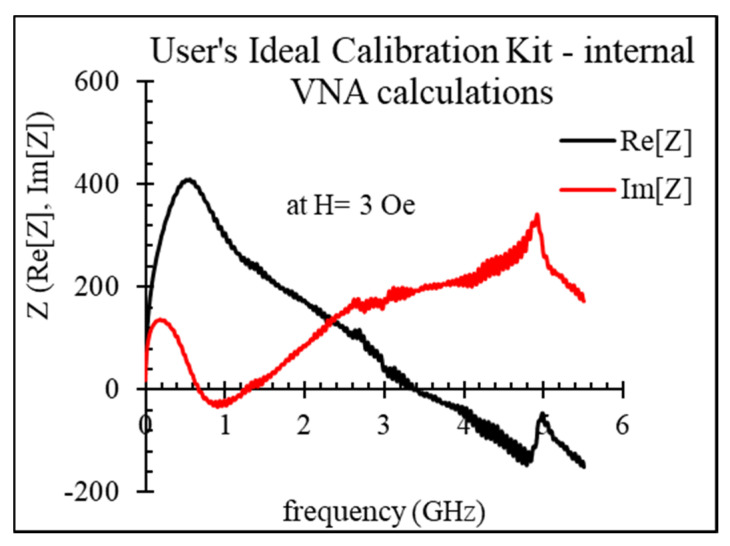
Dispersions of the impedance of amorphous Co_71_Fe_5_B_11_Si_10_Cr_3_ microwire (sample No2, as-cast) obtained from using user’s ideal calibration kit created on VNA.

**Figure 5 nanomaterials-11-01208-f005:**
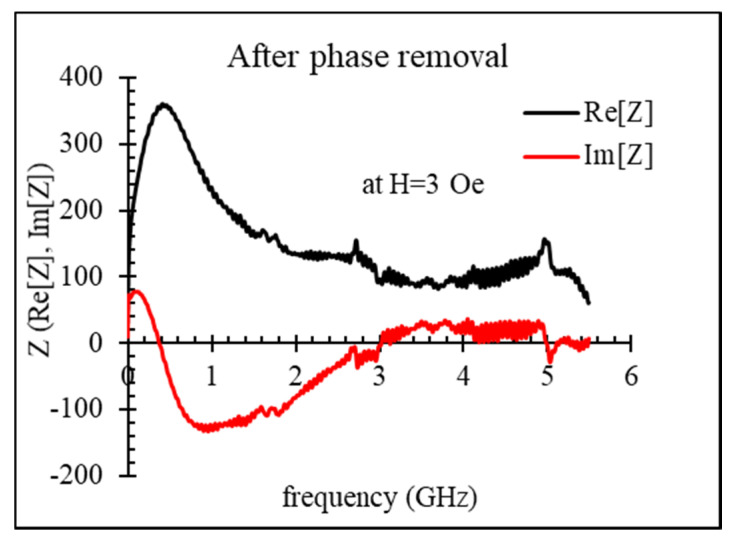
Dispersions of the impedance of amorphous Co_71_Fe_5_B_11_Si_10_Cr_3_ microwire (sample No2, as-cast) obtained from using user’s ideal calibration kit created on VNA after phase compensation.

**Figure 6 nanomaterials-11-01208-f006:**
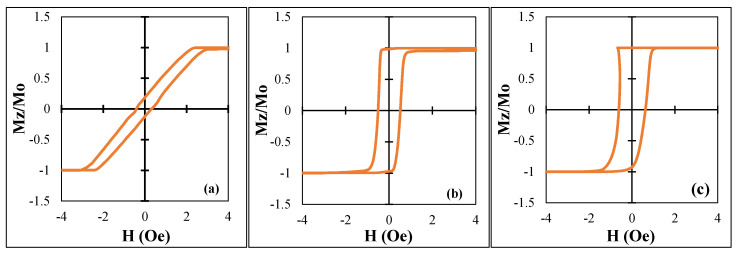
Hysteresis loops in a magnetic field along the wire axis for all the samples: (**a**) −1, (**b**) −2, and (**c**) −3.$\;M_{z}$ is the magnetization component along the wire, $M_{0}$ is the saturation magnetization.

**Figure 7 nanomaterials-11-01208-f007:**
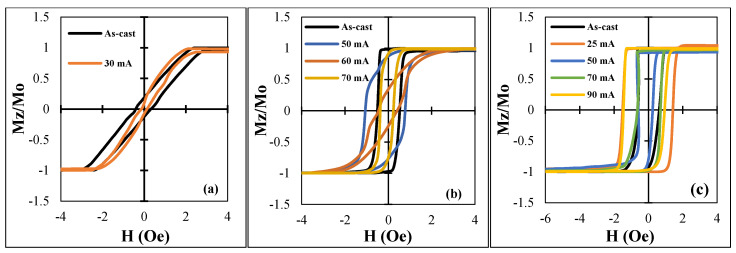
Hysteresis loops for all the samples after DC current annealing with different *I_an_* for 30 min: (**a**) −1, (**b**) −2, and (**c**) −3.

**Figure 8 nanomaterials-11-01208-f008:**
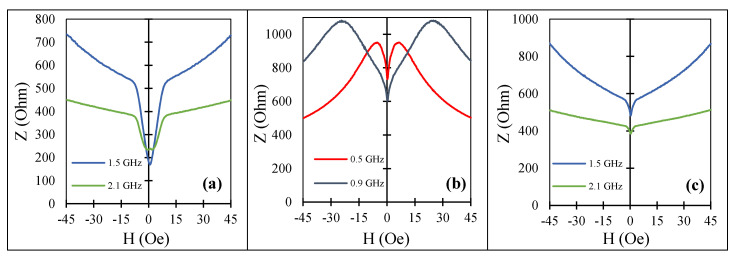
MI vs. magnetic field for amorphous Co_66.6_Fe_4.28_B_11.51_Si_14.48_Ni_1.44_Mo_1.69_ microwires (sample No1): (**a**) as-cast, (**b**,**c**) after current annealing with $I_{an}=30\;\mathrm{mA}$ for 30 min for different frequencies. Z corresponds to the impedance amplitude.

**Figure 9 nanomaterials-11-01208-f009:**
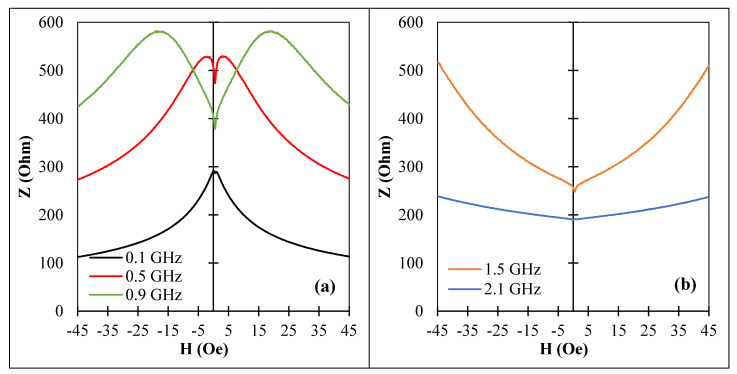
MI vs. magnetic field for amorphous Co_71_Fe_5_B_11_Si_10_Cr_3_ microwires (sample No2) in as-prepared state for different frequencies: (**a**) 0.1 GHz to 0.9 GHz and (**b**) 1.5 GHz to 2.1 GHz. Z corresponds to the impedance amplitude.

**Figure 10 nanomaterials-11-01208-f010:**
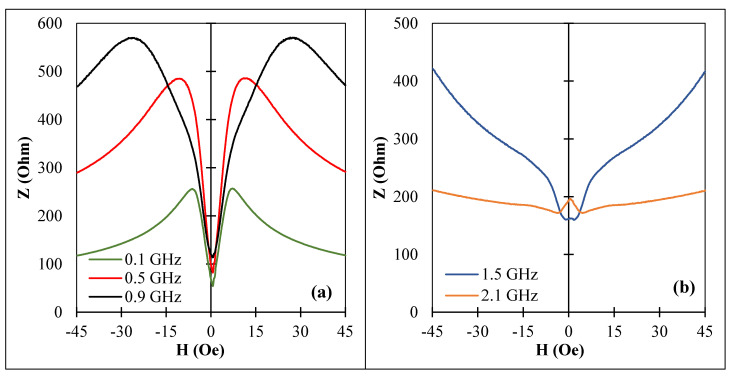
MI vs. magnetic field for amorphous Co_71_Fe_5_B_11_Si_10_Cr_3_ microwires (sample No2) after current annealing with $I_{an}=60\;\mathrm{mA}$ for 30 min at GHz frequency range: (**a**) 0.1 GHz to 0.9 GHz and (**b**) 1.5 GHz to 2.1 GHz. Z corresponds to the impedance amplitude.

**Figure 11 nanomaterials-11-01208-f011:**
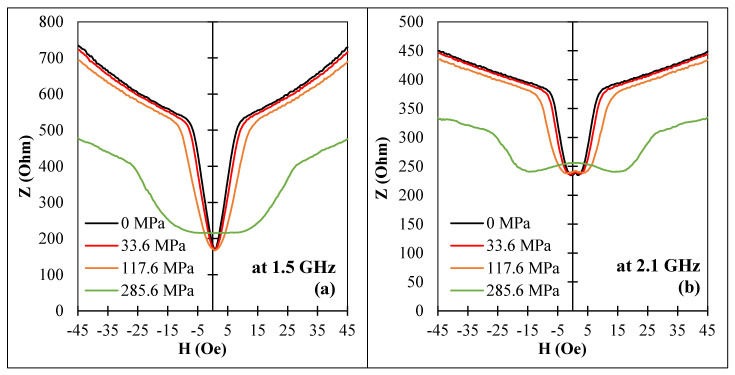
Effect of a tensile stress on the MI vs. magnetic field plots for amorphous Co_66.6_Fe_4.28_B_11.51_Si_14.48_Ni_1.44_Mo_1.69_ microwires (sample No1) in as-cast state at frequencies: (**a**) 1.5 GHz and (**b**) 2.1 GHz. Z corresponds to the impedance amplitude.

**Figure 12 nanomaterials-11-01208-f012:**
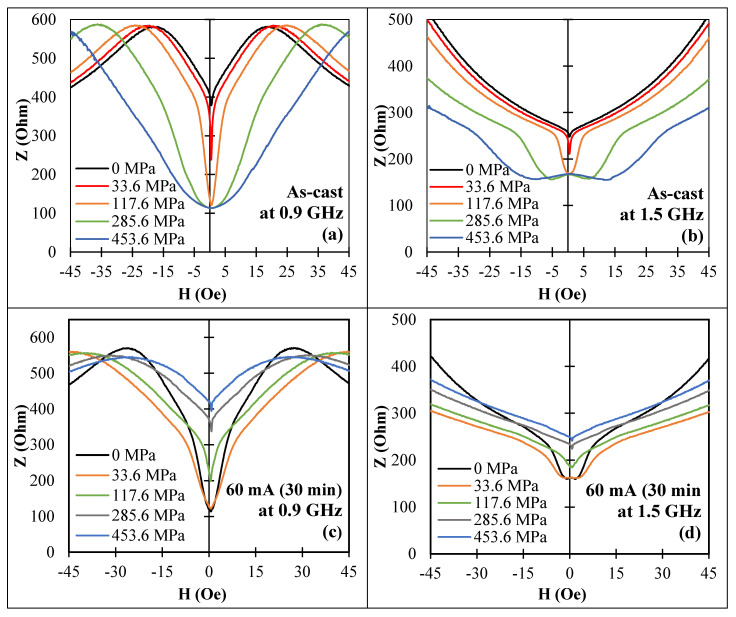
Effect of a tensile stress on MI vs. magnetic field for amorphous Co_71_Fe_5_B_11_Si_10_Cr_3_ microwires (sample No2) in as-cast state in (**a**,**b**) and after current annealing (60 mA, 30 min) in (**c**,**d**) for different frequencies: (**a**,**c**) −0.9 GHz; (**b**,**d**) −1.5 GHz. Z corresponds to the impedance amplitude.

**Figure 13 nanomaterials-11-01208-f013:**
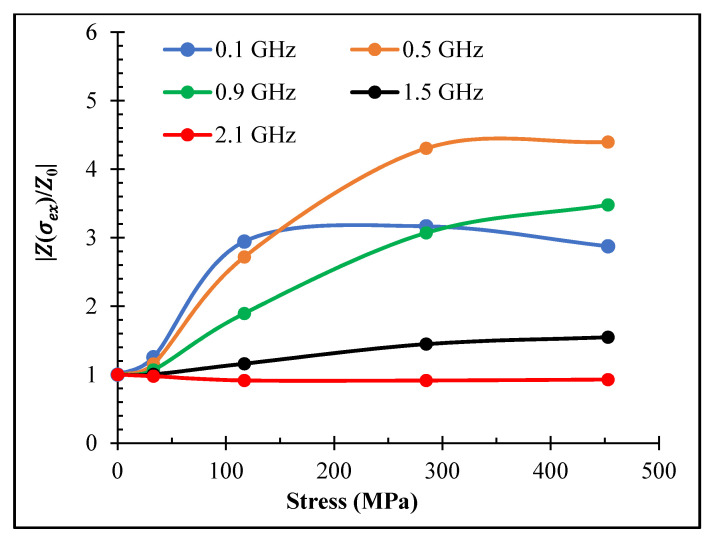
Stress dependence of MI at zero field *Z*(*H* = 0) for amorphous Co_71_Fe_5_B_11_Si_10_Cr_3_ microwires (sample No2) after DC current annealing at 60 mA for 30 min at different frequencies. $|Z\left(\sigma_{ex}\right)/Z_{0}\left|=\right|Z\left(H=0,\sigma_{ex}\right)/Z\left(H=0,\sigma_{ex}=0\right)$. $\sigma_{ex}$ is the applied stress.

**Figure 14 nanomaterials-11-01208-f014:**
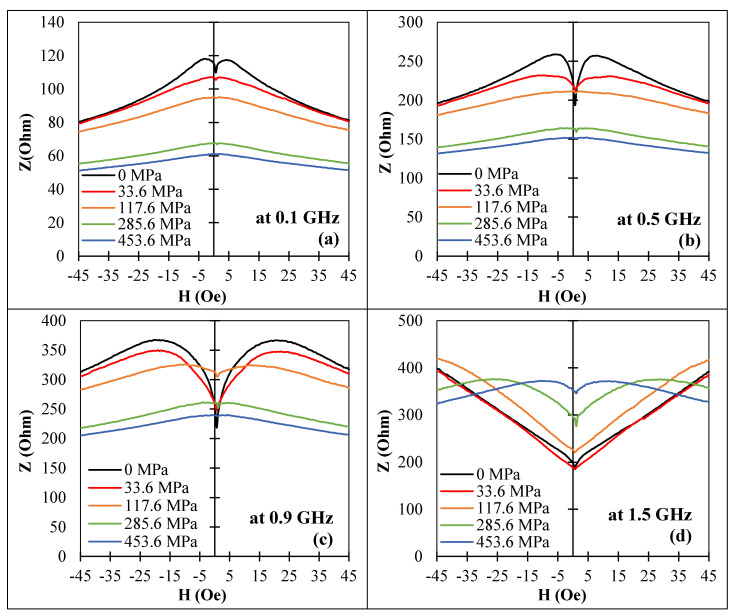
Effect of a tensile stress on the MI vs. magnetic field behavior for amorphous Co_71_Fe_5_B_11_Si_10_Cr_3_ microwires (sample No3) in as-prepared state at frequencies: (**a**) 0.1 GHz, (**b**) 0.5 GHz, (**c**) 0.9 GHz, and (**d**) 1.5 GHz. Z corresponds to the impedance amplitude.

**Figure 15 nanomaterials-11-01208-f015:**
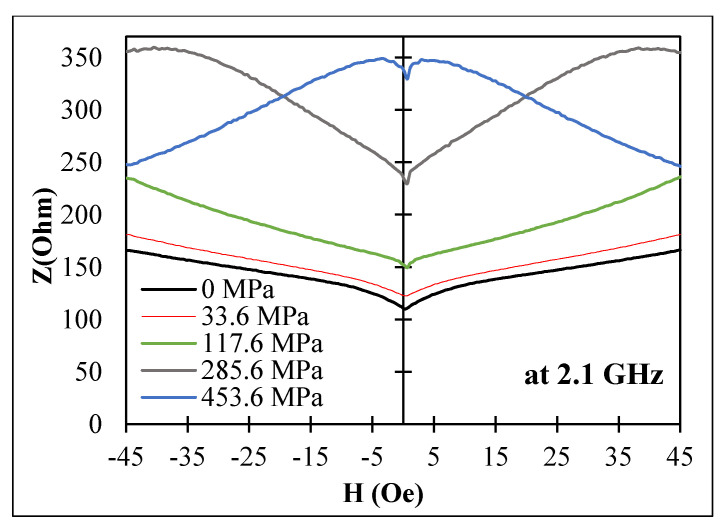
Effect of a tensile stress on the MI vs. magnetic field behavior for amorphous Co_71_Fe_5_B_11_Si_10_Cr_3_ microwires (sample No3) after DC current annealing at 90 mA for 30 min at a frequency of 2.1 GHz. Z corresponds to the impedance amplitude.

**Figure 16 nanomaterials-11-01208-f016:**
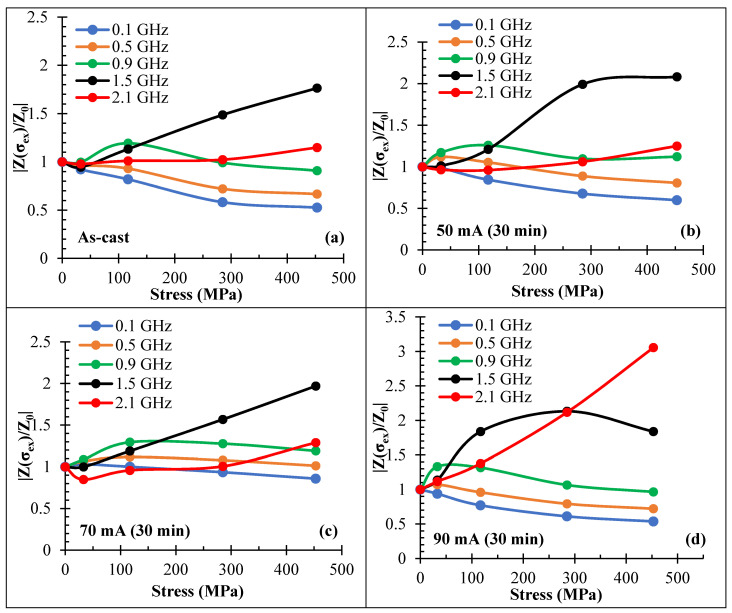
Stress dependence of MI at zero field (*Z*(*H* = 0)) for amorphous Co_71_Fe_5_B_11_Si_10_Cr_3_ microwires (sample No3) in as-prepared state (**a**) and after DC current annealing at 50 (**b**), 70 (**c**), and 90 (**d**) mA for 30 min at different frequencies. $|Z\left(\sigma_{ex}\right)/Z_{0}\left|=\right|Z\left(H=0,\sigma_{ex}\right)/Z\left(H=0,\sigma_{ex}=0\right)$. $\sigma_{ex}$ is the applied stress.

## Data Availability

The authors confirm that the data supporting the findings of this study are available within the article. Raw data were generated at differential scanning calorimetry, B-H loop meter and Hewlett-Packard 8753E Vector Network Analyzer. Derived data supporting the findings of this study are available from the corresponding author [L.P.] on request.

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
