# Peer review of "High-Frequency Magnetoimpedance (MI) and Stress-MI in Amorphous Microwires with Different Anisotropies"

_nanomaterials, 2021, doi:10.3390/nano11051208_

Round 1
Reviewer 1 Report
The manuscript “High Frequency Magnetoimpedance (MI) and stress-MI in Amorphous Microwires with Different Anisotropy” by J. Alam et al. is an important contribution to the field of sensors with the specific emphasis put on the high frequencies and its experimental assessment. It is a very good paper, logically organized and packed with important information exemplifying the main achievements. The manuscript is correctly organized and, I am confident, will be absorbed with interest by the community. So, concerning the merit of the manuscript I can hardly find any criticism. One solely comment would be a request the quality of the figures is low. I am not sure if it is due to the conversion to pdf, or because of the quality of the images. Moreover, you should be substituted “insert” by “inset” in the Figure 1.
Author Response
Comments and Suggestions for Authors:
We sincerely thank the reviewers for finding time to help us in improving the manuscript. We have made our best to properly address all the comments and suggestions. All corrections are highlighted in green.
Reviewers' comments and responses:
Reviewer #1
The manuscript “High Frequency Magnetoimpedance (MI) and stress-MI in Amorphous Microwires with Different Anisotropy” by J. Alam et al. is an important contribution to the field of sensors with the specific emphasis put on the high frequencies and its experimental assessment. It is a very good paper, logically organized and packed with important information exemplifying the main achievements. The manuscript is correctly organized and, I am confident, will be absorbed with interest by the community. So, concerning the merit of the manuscript I can hardly find any criticism. One solely comment would be a request the quality of the figures is low. I am not sure if it is due to the conversion to pdf, or because of the quality of the images. Moreover, you should be substituted “insert” by “inset” in the Figure 1.
Response:
The quality of figures has been improved, the caption in Figure 1 has also been corrected as suggested.

Reviewer 2 Report
The authors studied Magnetoimpedance and influence of tensile stress on magnetoimpedance for Co-based microwires. The paper is interesting and I recommend the publication of this paper in Nanomaterials.
However, an improvement of the work is necessary regarding the following comments.
1. Line 17 on page 1
The English may be corrected as:
used to characterized the microwave
-> used to characterize the microwave
2. In Eq. (1) sin and exp in line 168 on page 5 should be Roman characters.
3. Line 62 on page 2
The English may be corrected as:
proposed to established a required
-> proposed to establish a required
4. Line 117 on page 3, the authors wrote that "The procedure of current annealing and determination of the annealing temperate is given in 118 [20]."
In my opinion, it may be good that the author explains such a procedure briefly, if possible.
5. Line 137 on page 4, the authors wrote "printed circuit board (PCB)." However, the abbreviation "PCB" firstly appears in line 91 on page 3. Hence, the following revisions are necessary
(1) Line 91 on page 3: PCB -> printed circuit board (PCB)
(2) Line 137 on page 4: printed circuit board (PCB) -> PCB
6. \tilde{S}_{21} in line 170 on page 5 is not defined. The readers may not know the meaning of \tilde.
Author Response
Comments and Suggestions for Authors:
We sincerely thank the reviewers for finding time to help us in improving the manuscript. We have made our best to properly address all the comments and suggestions. All corrections are highlighted in green.
Reviewers' comments and responses:
Reviewer #2
The authors studied Magnetoimpedance and influence of tensile stress on magnetoimpedance for Co-based microwires. The paper is interesting and I recommend the publication of this paper in Nanomaterials.
However, an improvement of the work is necessary regarding the following comments.
- Line 17 on page 1
The English may be corrected as:
used to characterized the microwave
We changed this to “…employed to characterize the microwave…” (since the word “used” was used next to this part)
- In Eq. (2) sin and exp in line 168 on page 5 should be Roman characters.
All the equations are given using equation tool in word.docx
- Line 62 on page 2
The English may be corrected as:
proposed to established a required
This was changed accordingly.
- Line 117 on page 3, the authors wrote that "The procedure of current annealing and determination of the annealing temperate is given in [20]."
In my opinion, it may be good that the author explains such a procedure briefly, if possible.
The following short explanation was added
The wire samples were annealed by dc current in air atmosphere with the aim to investigate the effect of anisotropy and magnetostriction change on the MI and stress-MI behavior. The sample length for annealing was 15 cm, the dc current intensity was chosen between 25-90 mA and the flowing current time was 30 minutes. All current treatments were performed in the same ambient conditions. The current magnitude was chosen to realize a heating effect in the range of temperatures 450-750 K. For moderate annealing currents, the annealing temperature was measured using the temperature dependence of resistivity in a bridge circuit. For higher annealing currents (> 60mA) the annealing temperature was estimated from modelling on the basis of the energy balance. The details are given in [20].
- Line 137 on page 4, the authors wrote "printed circuit board (PCB)." However, the abbreviation "PCB" firstly appears in line 91 on page 3. Hence, the following revisions are necessary
(1) Line 91 on page 3: PCB -> printed circuit board (PCB)
(2) Line 137 on page 4: printed circuit board (PCB) -> PCB
The required changes were made: PCB abbreviation was explained at first appearance.
- \tilde{S}_{21} in line 170 on page 5 is not defined. The readers may not know the meaning of \tilde.
We clearly explained that was the wire lamped S21-parameter, which was found from the measured one after determining the time delay.

Reviewer 3 Report
The authors proposed an experimental study of high frequency (up to a few GHz) giant magnetoimpedance (GMI) in three type of microwires, with distinct magnetostrictive properties, each one submitted to different current annealing treatments. A general characterization of the GMI response as a function of field or frequency is presented, but the emphasis of the paper is on the effect of applied mechanical stress on the variation of the electrical impedance, under no applied magnetic field. The conclusion is that the large variation of the impedance as a function of stress could be exploited in sensing applications.
While the overall study is well done and suitable for publication in an appropriate journal with some minor corrections or clarification easy to implement, this reviewer has two main concerns:
- The link with the aims and scope of nanomaterials is not clear. The journal defines nanomaterials as materials with typical size features in the lower nanometer size range and characteristic mesoscopic properties; for example quantum size effects. One could argue that some of the microwires are nanocrystalline, although a little far-fetched, but GMI, a “classical” macroscopic electromagnetic effect based on skin effect. This hardly seem to qualify with the scope of a journal on nanomaterials.
- The novelty and significance of the work is not sufficiently highlighted. The effect of current annealing and of mechanical stress has been studied several times in the last 25 years, the underlying physics is well understood and, in spite of some possible marginal improvement in the calibration method, GMI measurements in the GHz range have been known for about 25 years also. The work is original and well done, but is there anything unexpected in the results to justify publication in nanomaterials?
Should the editor decide to go along with the publication, I could provide a list of specific comments and suggestion to the authors…
Author Response
Comments and Suggestions for Authors:
We sincerely thank the reviewers for finding time to help us in improving the manuscript. We have made our best to properly address all the comments and suggestions. All corrections are highlighted in green.
Reviewers' comments and responses:
Reviewer #3
The authors proposed an experimental study of high frequency (up to a few GHz) giant magnetoimpedance (GMI) in three type of microwires, with distinct magnetostrictive properties, each one submitted to different current annealing treatments. A general characterization of the GMI response as a function of field or frequency is presented, but the emphasis of the paper is on the effect of applied mechanical stress on the variation of the electrical impedance, under no applied magnetic field. The conclusion is that the large variation of the impedance as a function of stress could be exploited in sensing applications.
While the overall study is well done and suitable for publication in an appropriate journal with some minor corrections or clarification easy to implement, this reviewer has two main concerns:
- The link with the aims and scope of nanomaterials is not clear. The journal defines nanomaterials as materials with typical size features in the lower nanometer size range and characteristic mesoscopic properties; for example quantum size effects. One could argue that some of the microwires are nanocrystalline, although a little far-fetched, but GMI, a “classical” macroscopic electromagnetic effect based on skin effect. This hardly seem to qualify with the scope of a journal on nanomaterials.
- The novelty and significance of the work is not sufficiently highlighted. The effect of current annealing and of mechanical stress has been studied several times in the last 25 years, the underlying physics is well understood and, in spite of some possible marginal improvement in the calibration method, GMI measurements in the GHz range have been known for about 25 years also. The work is original and well done, but is there anything unexpected in the results to justify publication in nanomaterials?
Should the editor decide to go along with the publication, I could provide a list of specific comments and suggestion to the authors…
Response:
This manuscript was submitted to special issue-Novel Magnetic Properties in Curved Geometries. From this perspective and considering micro-size of materials used, the change in micromagnetic structure due to structural relaxation and atomic re-coordination, we believe that the manuscript suits the journal.
The novelty of manuscript is mainly related to realizing very high stress sensitivity of impedance change at GHz frequencies without using any bias field and physical explanation of this effect. The work also clearly demonstrates the difference in two mechanisms of magnetoimpedance effect: due to dc magnetization re-orientation and due to change in axial magnetic anisotropy. The latter configuration has not received attention since it was considered unfavorable for MI. However, at high frequencies the change in axial easy anisotropy produces very large and sensitive change in impedance. We also demonstrated that enhancing the magnetoelastic anisotropy by annealing partially crystalline wires makes it possible to progressively increase the frequency of high stress-impedance change.
We made additional emphasis for novelty in the abstract and introduction.

Round 2
Reviewer 3 Report
This reviewed version better emphasizes the novelty and significance of the work (Point 2). As for Point 1, if the editor finds the work suitable for the scope of Nanomaterials, this reviewer certainly will not object. A certain number of misprints were already corrected and the English was improved. Therefore I recommend its publication.
Nevertheless, I invite the authors to consider a few small comments and suggestions as outlined below:
-Section 2.1: It would be relevant to mention the sign of the magnetostriction and expected as-cast anisotropy of the different types of wires this early in the paper.
-Section 2.2: The direction of the applied field should be mentioned in the text for clarity (I saw that it was added in a caption, which is better than nothing).
-Figure 5 interestingly exhibited two resonances: It is probably worth mentioning it at least, and elaborating on it if the authors have an explanation.
-Figure 6c) exhibit some asymmetry: it is probably worth mentioning, is it possibly associated with some helical anisotropy? Or other explanation?
-Section 3.2: the authors should clarify their definition of impedance ratio, as there are different definitions used in the literature (reference impedance is at different fields).
-line 278: it would seem that the optimal current annealing conditions are not specified…
-line 304 – 306: Sample 2 is claimed to have negative magnetostriction before annealing (line 304) and then positive magnetostriction after annealing (line 305). Is that a misprint or the annealing changed the magnetostriction? If so, that would be quite odd, isn’t it? Please elaborate…
-Line 58: the high frequency GMI methodology was initially developed independently in Moscow and Montreal and presented in J. Appl. Phys. 81 (8), 4032 (1997). It would seem fair to this reviewer to point out to that work, or to article with very similar figure as the authors Fig 1, such as in J. Appl. Phys., Vol. 83, No. 11, p. 6563 (1998). This is of course at the discretion of the authors.
Author Response
Comments and Suggestions for Authors:
We sincerely thank the reviewers for finding time to help us in improving the manuscript. We have made our best to properly address all the comments and suggestions. All corrections are highlighted in yellow.
Reviewers' comments and responses:
This reviewed version better emphasizes the novelty and significance of the work (Point 2). As for Point 1, if the editor finds the work suitable for the scope of Nanomaterials, this reviewer certainly will not object. A certain number of misprints were already corrected and the English was improved. Therefore I recommend its publication.
-Section 2.1: It would be relevant to mention the sign of the magnetostriction and expected as-cast anisotropy of the different types of wires this early in the paper.
- Information about the magnetostriction constant and anisotropy type of as-cast samples was added.
Co-based wires typically have a negative magnetostriction and a circular type of the easy anisotropy since the internal stress is predominantly axial. Samples No 1 with a larger ratio of Co/Fe~15 content had the magnetostriction constant of about [30, 33] and a circumferential easy anisotropy. Sample No 2 had a smaller negative magnetostriction (< [21, 34]) but an axial easy anisotropy. Sample No 3 had a large and positive magnetostriction constant of about due to partial crystallization [34] and an axial easy anisotropy.
-Section 2.2: The direction of the applied field should be mentioned in the text for clarity (I saw that it was added in a caption, which is better than nothing).
- The information about the field direction was added.
For hysteresis loop measurements an AC magnetic field of a frequency of 500 Hz was used. The direction of external field was along the wire length (axial direction).
-Figure 5 interestingly exhibited two resonances: It is probably worth mentioning it at least and elaborating on it if the authors have an explanation.
- The following explanation was added.
The resonance seen at a frequency of about 5 GHz is a parasitic electric resonance related to the OPEN terminations on the PCB calibration cell. It cannot be removed, and a sort of smooth interpolation is needed.
-Figure 6c) exhibit some asymmetry: it is probably worth mentioning, is it possibly associated with some helical anisotropy? Or other explanation?
- The loop appears to be asymmetrical, but this is related to digital integration of a sharp voltage pulse induced during re-magnetization. In this case, sampling needs to be done with higher frequency which was not possible within our BH-loop tracer. This explanation was added.
-Section 3.2: the authors should clarify their definition of impedance ratio, as there are different definitions used in the literature (reference impedance is at different fields).
- The definition of the impedance ratio was added.
The impedance change is determined with respect to its value at zero field and zero stress.
-line 278: it would seem that the optimal current annealing conditions are not specified…
The purpose of this work was to investigate the high frequency impedance for different combinations of easy anisotropy and magnetostriction which was achieved using different samples (content and geometry) and different annealing conditions. The determination of the optimal conditions of current annealing should be done for specific applications. For example, in order to obtain high stress-impedance at zero field in the GHz frequency range, a wire with relatively large and positive magnetostriction is needed. This could be achieved by annealing with higher intensity (90 mA) of partially crystalline wire as demonstrated in this work.
-line 304 – 306: Sample 2 is claimed to have negative magnetostriction before annealing (line 304) and then positive magnetostriction after annealing (line 305). Is that a misprint or the annealing changed the magnetostriction? If so, that would be quite odd, isn’t it? Please elaborate…
- The effect of current annealing of Co-based amorphous samples with small and negative magnetostriction leads to change in its sign and absolute value. This is due to structural relaxation and change in atomic co-ordination. This explanation with references [21, 36] was added. Reference 36 is new.
-Line 58: the high frequency GMI methodology was initially developed independently in Moscow and Montreal and presented in J. Appl. Phys. 81 (8), 4032 (1997). It would seem fair to this reviewer to point out to that work, or to article with very similar figure as the authors Fig 1, such as in J. Appl. Phys., Vol. 83, No. 11, p. 6563 (1998). This is of course at the discretion of the authors.
We agree with this comment and an appropriate reference was added (present reference [14])
Ciureanu, P.; Britel, M.; Ménard, D.; Yelon, A.; Akyel, C.; Rouabhi, M.; Cochrane, R.W.; Rudkowski, P.; Ström-Olsen, J.O. High frequency behavior of soft magnetic wires using the giant magnetoimpedance effect. J. Appl. Phys. 1998, 83, 6563–6565, doi:10.1063/1.367602.
